# KnowSelf: Efficicent Knowledge Retrieving for Legal Reasoning

## Abstract

Large language models (LLMs) have made significant strides in the legal domain, such as recommending charges based on the given criminal fact. However, existing LLMs struggle to address the long-tail charge prediction problem. This is primarily due to the imbalanced distribution of legal data during pre-training, resulting in varied inference capabilities, especially those in the long-tail accusation category. Moreover, common methods for enhancing LLMs' reasoning abilities, such as the chain-of-thought series and retrieval-augmented generation series, also fail to address the long-tail charge prediction problem. In this work, we reveal that solving this issue requires providing LLMs with their deficient legal knowledge. We propose a legal knowledge retrieval method (denoted as KnowSelf) that includes a *Knowledge Inspector* to identify their knowledge gaps and a *Knowledge Integrator* to provide tailored legal knowledge for accurate legal reasoning. Extensive experiments on real-world datasets demonstrate that our method significantly surpasses prior state-of-the-art methods, e.g., achieving average F1 gains of 22.91% for overall charges and 22.94% for tail charges on GPT-4o, and gains of 26.74% (overall) and 27.16% (tail) on QWen2.5-7B. Our codes and data are available at Github: `https://anonymous.4open.science/r/KnowSelf-F860`.

## 1 INTRODUCTION

The charge prediction task aims to recommend a charge based on given criminal facts, which serves as a fundamental problem in the field of Legal Artificial Intelligence (LegalAI) (Ye & Li, 2024; Qin et al., 2024). Today, there is an increasing demand for reliable charge prediction models that can enhance the fairness of legal decision-making, benefiting both professionals and the public by improving accessibility and promoting equity in the judiciary.

Recent advancements in Large Language Models (LLMs), such as GPT-4 (OpenAI, 2023), Deepseek (DeepSeek-AI et al., 2025), and Qwen (Qwen et al., 2025), have demonstrated remarkable performance across various domains (Mao et al., 2024), including LegalAI (Shui et al., 2023; Wei et al., 2024). To improve LLMs' reasoning capabilities and bring them closer to human-like cognition, Chain-of-Thought (CoT) (Wei et al., 2022) has been developed to encourage step-by-step reasoning. Additionally, domain-specific LLMs, e.g., ChatLaw (Cui et al., 2023), are trained on extensive legal documents and tasks, significantly improving legal knowledge and expertise. Consequently, with implicitly stored knowledge and the emergent capabilities of large models, LLM-based methods have become an outstanding approach for charge prediction.

Despite great improvements, LLMs still struggle to predict long-tailed charges. In practice, charges do not appear equally often; many are rarely represented or occurred. As shown in Fig. 1 (a), the statistical analysis of large-scale legal cases from *China Judgment Online* [1] (CJO), an open access goverment website, reveals a significant long-tailed distribution, with over 88% of charges accounting for less than 0.5% of the total cases. This imbalanced distribution of legal domain information limits the reasoning ability of LLMs when addressing long-tailed charges. For instance, GPT-4o achieves only 33% accuracy for tail charge prediction, with a 29% decrease compared to head charge predictions; while this issue is even more evident in smaller LLMs like QWen2.5-7B, where tail charge accuracy drops to just 28.14%.

---

[1] `https://wenshu.court.gov.cn/`

To tackle the long-tailed problem, a well-known solution involves retrieval-augmented generation (RAG) methods (Xu et al., 2024; Zhou et al., 2024; Salemi & Zamani, 2024), which enhance LLM-generated responses by integrating relevant information through retrieval. For instance, Kandpal et al. (2023b) employ entity linking to search relevant documents for LLM-based question answering. Li et al. (2024a) retrieve documents to assist LLMs when user queries relate to long-tail knowledge. In LegalAI, RAG methods (Deng et al., 2024; Tang et al., 2024b; Pipitone & Alami, 2024) are also applied to improve

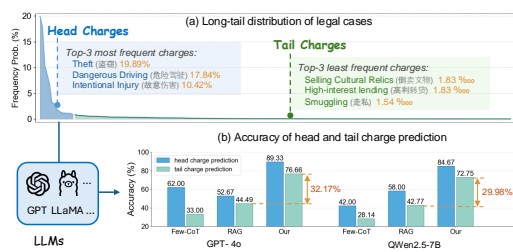

Figure 1: Illustration of (a) Long-tail distribution of charges on CJO and (b) Accuracy of head and tail charge predictions using GPT-4o and QWen-2 on CAIL (Xiao et al., 2018).

LLMs' legal understanding and reasoning abilities through *legal case retrieval* (Ye & Li, 2024; Deng et al., 2024; Tang et al., 2024a). However, in practice, it is observed that RAG systems fall short when addressing long-tail charge prediction. As shown in Fig. 1 (b), RAG on GPT-4o leads to a performance drop for tail charges, revealing their limitations.

The shortcomings of existing RAG methods in LegalAI have been widely explored (Barnett et al., 2024; Magesh et al., 2024). The primary reason is that legal case retrieval is particularly challenging (Mik, 2023; Magesh et al., 2024); *relevance* in the legal context is not based on text alone, whereas most retrieval systems identify relevance based primarily on *textual similarity* (Karpukhin et al., 2020; Liu et al., 2023b; Cuconasu et al., 2024). As shown in Fig. 2, given a legal case that should be judged as the crime of `Privately Carving up State-owned Property`, the retrieved two cases are textually similar but are legally unrelated —specifically, cases of `Giving Bribery` and `Corruption`—which further misleads the LLMs. Intuitively, legal case retrieval becomes even more difficult when addressing long-tailed charges, as relevant cases are scarce in large-scale candidate cases, and introducing irrelevant cases can negatively impact the effectiveness of LLMs.

In this work, we reveal the importance of *legal knowledge retrieval* for long-tail charge prediction and propose a new method named Know-Self. The motivation is to retrieve legal knowledge that LLMs have not learned adequately, by providing comparative legal insights to help distinguish between tail charges and commonly confused charges. Specifically, we first design a knowledge inspector module to identify the missed long-tail knowledge for LLMs. The key intuition driving this design is that legal professionals with varying legal backgrounds need to consult relevant resources when addressing legal tasks beyond their expertise. For LLMs, revealing legal knowledge gaps is crucial for improving their legal reasoning capabilities. Then, we design a knowledge integrator module to provide legal insights of charges retrieved based on the identified legal knowledge gaps of the LLM. Finally, we strengthen the legal reasoning abilities of LLMs by compensating for their gaps in legal knowledge.

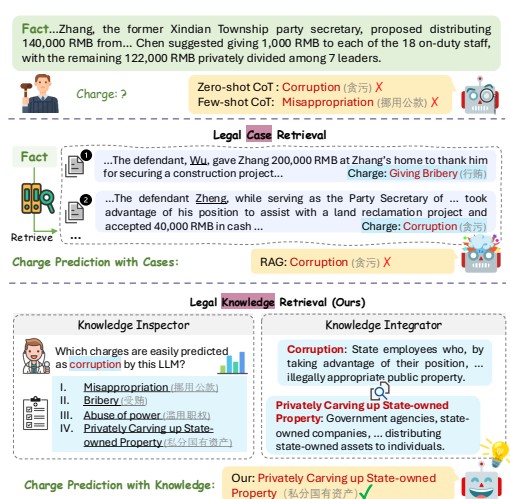

Figure 2: Illustration of charge prediction using case retrieval and our knowledge retrieval.

Overall, our method KnowSelf is advanced in improving long-tail charge prediction, by reducing the bias toward head charges resulting from imbalanced training data. Moreover, unlike legal case retrieval which typically provides surface-level cases as background with the risks of negative distractions, our long-tailed legal knowledge offers a deeper and more reliable foundation for enhancing the understanding of tail charges.

Extensive experiments show that KnowSelf significantly surpasses the previous state-of-the-art (SOTA) methods by improving 19.78% (overall) and 20.07% (tail) in terms of the average F1.

To enable future research, we release our code and data at `https://anonymous.4open.science/r/KnowSelf-F860`. The contributions are summarized as follows:

- We emphasize the importance of improving long-tailed charge prediction with LLMs, showing that legal knowledge retrieval is more effective than legal case retrieval in addressing this issue. To the best of our knowledge, we are the first to integrate long-tail legal knowledge to reduce LLM biases in LegalAI.

- We design a novel legal knowledge retrieval method for LLMs, which includes a knowledge inspector to identify long-tailed knowledge and a knowledge integrator module to incorporate this knowledge into LLMs.

- Extensive experiments show our method's superior capabilities, achieving average F1 improvements of 16.81% (overall) and 22.6% (tail) on GPT-4o-mini and GPT-4o, as well as 16.09% (overall) and 18.39% (tail) on small-scale LLMs like QWen2.5-1.5B, and QWen2.5-7B, effectively mitigating LLM biases in inference.

## 2 PRELIMINARY

### 2.1 TASK DEFINITION

**Charge Prediction** This task involves predicting the appropriate criminal charge based on the fact description of a legal case. Formally, a *fact* refers to the description of a criminal case and is represented as $f = \{w_1, w_2, ..., w_{|f|}\}$, where the $w_i$ is the $i$-th word. The charge label set is denoted as $\mathcal{Y} = \{y_1, y_2, \dots\}$, where each $y_i$ corresponds to a legally predefined charge, such as `Intentional Homicide`, `Theft`, and `Intentional Injury`. Given the fact of a legal case $f$, the goal of this task is to predict the applicable charge $y$ from $\mathcal{Y}$.

**Head and Tail Accusations** For $\mathcal{Y}$, charges are categorized into two types, i.e., *head* charges and *tail* charges, based on their frequency in legal cases collected from CJO. Specifically, all charges are ranked by frequency, and the top-$n$ charges that constitute 80% of the total cases are termed *head* charges, with the remainder classified as *tail* charges. There are 15 head charges and 172 tail charges in $\mathcal{Y}$. Appendix A details the statistical information and summarizes several head and tail charges and their frequency probabilities.

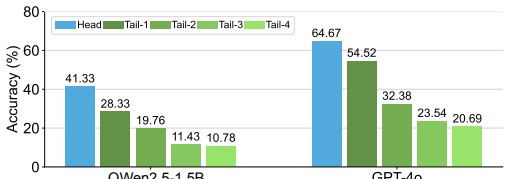

Figure 3: Comparison of head and tail charge prediction. The X-axis represents charges, with one head group and four tail groups arranged from high to low frequency.

### 2.2 PROBLEMS OF LONG-TAIL ACCUSATION PREDICTION

Recently, LLM-based methods have become mainstream for charge prediction, relying on their deep contextual understanding of legal texts and extensive background knowledge stored through large-scale pre-training. Although these methods have achieved notable results, they remain insufficient in effectively addressing the long-tail problem (Kandpal et al., 2023a). To illustrate this problem, we conducted probing experiments and in-depth analysis.

First, tail charge prediction exhibits low performance. Fig. 3 illustrates the performance of head and tail charges for both LLMs, confirming that infrequent charges tend to show poorer performance. Second, tail charge prediction can be easily distracted by certain charges. Fig. 10 shows the LLMs' prediction biases for all mispredictions. Both the X-axis and Y-axis are arranged from high to low frequency of charges, with the X-axis representing true charges $y$ and the Y-axis representing predicted charges $\hat{y}$. Larger, darker circles indicate a higher frequency of misclassification, suggesting that LLM struggles to distinguish between specific charges[2]. As Fig. 10 shows, the circles are concentrated at the lower right corner of the figure, which can be seen as concrete evidence of LLM's tendency to be distracted by head charges while performing tail charge prediction. For instance, we

---

[2]Ideally, correct predictions would be concentrated along the diagonal (where the true label and predicted label match), but we have omitted these instances for clarity.

observed that the tail charge `Assisting in destroying or fabricating evidence` are frequently predicted as `Dangerous Driving` and `Traffic accident offense`. This motivates us to inject long-tail knowledge into LLMs to reduce confusion between tail charges and specific charges that are often misclassified.

## 2.3 LIMITATIONS OF LEGAL CASE RETRIEVAL

Recently, legal case retrieval (Shui et al., 2023; Wei et al., 2024) has garnered significant attention for its potential to enhance LLM performance by providing similar legal cases. However, retrieving legal cases only based on textual similarity is not enough, since distinct charges may share similar criminal facts (Mik, 2023; Magesh et al., 2024; Deng et al., 2024; Li et al., 2024c). For example, both the charges `Intentional Injury` and `Intentional Homicide` are involved in violence, injury, and even death. These issues limit the performance of legal case retrieval methods (Gao & Callan, 2021; 2022; Lu et al., 2021; Liu et al., 2023c), as detailed in Section 4.6 and Table 3.

Unlike *legal case retrieval*, which aims to augment LLMs with useful cases, our *legal knowledge retrieval* emphasizes identifying the gaps or biases in the legal knowledge within LLMs, and then providing missing and useful legal knowledge. The advantages of our *legal knowledge retrieval* are summarized as follows:

- Our method provides precise knowledge for LLMs, whereas legal case retrieval may present unrelated cases and increase distractions.

- We offer deep knowledge specifically for tail charges, making it easier to distinguish between similar charges, rather than superficial legal cases.

- Different LLMs have learned varying legal knowledge, and our method provides tailored knowledge for each. In contrast, legal case retrieval uses fixed retrievers to supply the same cases to all LLMs, overlooking their differences.

## 3 METHODOLOGY

The overall architecture of our method KnowSelf is shown in Fig. 4. Specifically, the *Knowledge Inspector* module is designed to assess which types of legal knowledge the LLM is proficient in. Following this, the *Knowledge Integrator* module retrieves the missing legal knowledge according to the findings of the *Knowledge Integrator* module. Lastly, the *Legal Reasoner* module conducts legal reasoning based on the integrated knowledge. We detail the three modules below.

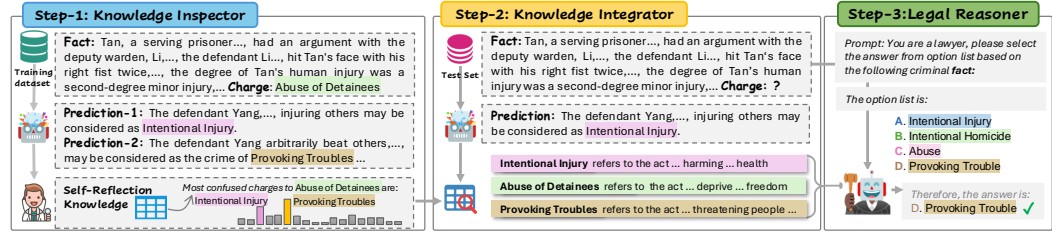

Figure 4: Overall architecture of our method *KwowTail* including the *Knowledge Inspector* module, *Knowledge Integrator* module, and the *Legal Reasoner* module. In particular, we first use the *Knowledge Inspector* to get the self-reflection knowledge for the given LLM. Then, for the given legal case, the *Knowledge Integrator* is used to retrieve the knowledge the LLM requires. Finally, *Legal Reasoner* elicits the LLMs to conduct legal reasoning by providing the retrieved knowledge.

## 3.1 KNOWLEDGE INSPECTOR

In practice, legal professionals possess specialized fields of law, such as business law or criminal law. When faced with unfamiliar fields, they need to acquire new legal knowledge and augment their understanding. Likewise, LLMs need to access pertinent knowledge when performing tasks outside their areas of expertise. The *Knowledge Inspector* module serves to detect the categories of legal knowledge that the LLM is deficient in. Specifically, for the given LLM and each case within the training dataset $\mathcal{D}_{train} = \{(f, y)_i\}_{i=1}^n$,

we instruct the LLM to reason the charge label under the zero-shot setting. The prompting template is shown in Fig. 5, which begins with a question to elicit the LLM to predict the charge $\hat{y}_i$. Inevitably, the output of the LLMs may be out of the charge label set $\mathcal{Y}$. To tackle this issue, we select the top-$s$ charge labels from the $\mathcal{Y}$ that are similar to the predicted charge label as the reasoning results by using the BM25 algorithm [3]. Finally, for each case $f_i \in \mathcal{D}_{train}$, we obtain its paired predicted and ground truth charge labels, denoted as $\mathcal{P} = \{(f, \hat{y}, y)_i\}_{i=1}^n$. For each paired

**Zero-Shot Charge Reasoning**

You are a lawyer, please predict the charge based on the following criminal fact:

**Criminal Fact:** it was found that ..., the defendant Wu went to the classroom of ...College and stole a rose gold VIVO brand X9 mobile phone that the victim Gao had stored in the mobile phone storage box in the classroom. It was determined that the stolen mobile phone was worth RMB 2,289. It was also found that the defendant Wu truthfully confessed ....

**Answer:**

Figure 5: The template of prompt LLMs for charge reasoning under the zero-shot setting.

charge, if $y_i \neq \hat{y}_i$, we argue that the given LLM is susceptible to confusion between the legal principles $y_i$ and $\hat{y}_i$. Next, we construct a table $T$ based on statistics information of these paired charge labels in $\mathcal{P}$. The element $t_{i,j}$ of table $T$ denotes the probability that LLM predicts the correct charge $y_i$ as the wrong charge $y_j$. We depict the table $T$ in Fig. 10. It is observed that most of the LLMs' wrong-predicted charge labels follow a long-tailed distribution, which is denoted as the self-reflection knowledge in this work. It indicates that more attention should be paid to the knowledge of the tail labels which the LLM is deficient in.

## 3.2 KNOWLEDGE INTEGRATOR

For the same case, LLMs with different capacities necessitate distinct legal knowledge. To accommodate this, based on Table $T$, the *knowledge integrator* module aims to mind the knowledge gaps in LLMs. Specifically, for a test legal case $f_j$, we first obtain the charge label $y_j$ by conducting zero-shot charge reasoning using the prompt template shown in Fig. 5. Then, based on the table $T$, we select the top-$k$ charge labels (excluding $y_j$) that are most prone to being misjudged as

**Charge Label Explain**

You are a lawyer, please explain the charge labels based on the Criminal Law of China:

**Theft:** It refers to the act of secretly stealing a large amount of public or private property or stealing public or private property multiple times for the purpose of illegal possession.

**Intentional injury:**

Figure 6: The template of prompt LLMs to explain the charge labels.

charge $y_j$ by the LLM as candidate labels, denoted as $\mathcal{C}_j$. Then, the LLM should explain each label in the $\mathcal{C}_j$. The template is shown in Fig. 6, which begins with a question and demonstration. Finally, the LLM needs to generate an explanation of the given charge labels, which is denoted as legal rule knowledge in the following sections.

## 3.3 LEGAL REASONER

option from the following list. Besides, we construct a reasoning path to elicit the LLM to generate the answer, which mainly consists of the explanation of options. After retrieving the knowledge required for handling given legal cases, the legal reasoner module aims to prompt the LLM to make a decision, based on the retrieved knowledge. The prompting template is shown in Fig. 7. The template begins with instructions, asking the LLM to select an After retrieving the knowledge required for handling given legal cases, the legal reasoner module aims to prompt the LLM to

**Legal Reasoner**

You are a lawyer, please select the answer from option list based on the following criminal fact:

**Criminal Fact:** ..., the defendant Wang took advantage of Ma's sleep and slashed the victim's neck and stabbed her face with a knife. ..., Ma's injury was grade II severe injury. According to.....

**Options:** A. Intentional Injury  B. Intentional Homicide  C. Abuse  D. Provoking Trouble

**Response:** The option A refers to acts that intentionally and illegally harm the physical health of others; The option B refers to ...; The option C refers to ...; The option D refers to .... Therefore, the answer option is:

Figure 7: The prompt template for charge reasoning based on the retrieved knowledge.

make a decision, based on the retrieved knowledge. The prompting template is shown in Fig. 7. The template begins with instructions, asking the LLM to select an option from the following options list. Besides, we construct a reasoning path to elicit the LLM to generate the answer, which mainly consists of the explanation of options.

---

[3]https://pypi.org/project/rank-bm25/

# 4 EXPERIMENT AND SETTINGS

To evaluate the effectiveness of our KnowSelf, we conduct comprehensive experiments to answer the following research questions: **RQ-1**: How effective is our method in long-tail charge prediction? **RQ-2**: What role does legal knowledge play? **RQ-3**: Do legal cases matter? **RQ-4**: How does our method perform on a specific case?

## 4.1 LLMs

We first introduce the used LLMs. **QWen2.5** (Qwen et al., 2025) is an open-sourced LLM, pre-trained on a stable dataset comprising up to 3 trillion tokens of multilingual data, spanning a broad range of domains. The versions of `QWen2.5-7B` are used in our study [4]. **DeepSeek** (DeepSeek-AI et al., 2025) is an open-source pre-trained LLM with 671 billion parameters [5]. **GPT-4o** is available from OpenAI API and the versions of `GPT-4o-2024-08-06` are used [6].

## 4.2 DATASET AND EVALUATION METRICS

In this study, we conduct experiments on the widely used Chinese legal judgment prediction datasets CAIL-Small and CAIL-Large (Xiao et al., 2018). Each instance consists of the criminal fact and the corresponding labeled charge. Following the previous study (Shui et al., 2023), we sample a balanced small training set and test set from the original dataset, respectively. Specifically, for each charge label, we randomly sample ten criminal cases. The small training set is used to construct the self-reflection knowledge. Following Shui et al. (2023), we use the BM25 algorithm to measure the similarity between legal cases, and the performance of other retrievers is shown in Table 3. We employ the accuracy to evaluate the capability of baselines and our method.

## 4.3 EXPERIMENT SETTING

**Baselines.** We compare our method against two groups of baselines. (1) Fine-tune-based Methods, which involve fine-tuning pre-trained language models, including BERT (Devlin et al., 2019) and RoBERTa (Cui et al., 2021) (both are Chinese versions); and Lawformer (Xiao et al., 2021) which is pre-trained specifically on legal domain; (2) Prompt-based Methods, CoT Kojima et al. (2022); Wei et al. (2022), Self-Consistency (Wang et al., 2023) which elicits LLMs think step by step and obtain finall answer by major-voting; and RAG (Shui et al., 2023; Wei et al., 2024), which enhances LLMs with similar cases as demonstration.

**Implementation Details.** We implement baselines using the released source codes. For fine-tune-based methods, we set batch size, learning rate, dropout rate, warmup steps, and max length of criminal facts as 16, $1 \times 10^{-5}$, 0.1, 200, and 500, respectively. We fine-tune these models on the Tesla A100 40GB*1 GPU with the AdamW Loshchilov & Hutter (2019) optimizer for 10 epochs. For prompt-based methods and our method, following the previous study Shui et al. (2023), we use the BM25 algorithm [7] to implement similar case retrieval and map LLMs' outputs to predefined charge labels. For our method, we set the top-$s$ (used for the knowledge inspector module) as 2 and the top-$k$ (used for the knowledge integrator module) range from 3 to 7. All experiments are conducted 3 times with distinct random seeds. We report the best average performance on the sampled test set.

## 4.4 PERFORMANCE ON ACCUSATION PREDICTION (RQ-1)

The overall performance is shown in Table 1. It is observed that, for prompting methods, **(1)** our method significantly outperforms the previous SOTA methods, achieving an average 19.78% improvement with different LLMs. **(2)** Our method achieves solid improvements on tail charge prediction. For instance, on QWen2.5-7B and GPT-4o, our method outperforms previous SOTA methods,

---

[4]https://huggingface.co/Qwen

[5]https://www.deepseek.com/

[6]https://openai.com/

[7]https://pypi.org/project/rank-bm25/

| | | Fine-Tuning | | | CoT | | | Self-Consistency | | | GAG | | | **KnowSelf** | | |
|---|---|---|---|---|---|---|---|---|---|---|---|---|---|---|---|---|
| | | BERT | RoBERTa | Former | Qwen | R1 | GPT | Qwen | R1 | GPT | Qwen | R1 | GPT | Qwen | R1 | GPT |
| **Head** | CAIL-S | 80.01 | 82.07 | 81.13 | 64.03 | 66.45 | 63.89 | 65.49 | 67.80 | 65.23 | 58.78 | 63.90 | 63.02 | 84.67 | **90.12** | 89.33 |
| | CAIL-L | 81.22 | 83.00 | 82.78 | 63.45 | 67.00 | 63.70 | 63.99 | 69.33 | 63.82 | 62.25 | 62.83 | 63.67 | 85.15 | **89.17** | 89.50 |
| **Tail** | CAIL-S | 77.60 | 79.01 | 79.15 | 38.03 | 40.45 | 48.90 | 38.64 | 41.01 | 38.94 | 42.75 | 52.69 | 59.24 | 72.60 | **79.78** | 76.66 |
| | CAIL-L | 75.83 | 76.24 | 78.03 | 35.65 | 37.00 | 41.33 | 35.77 | 38.68 | 42.54 | 44.34 | 56.87 | 58.89 | 73.50 | **80.01** | 75.83 |
| **All** | CAIL-S | 77.78 | 79.10 | 79.33 | 47.15 | 59.30 | 60.88 | 46.56 | 61.70 | 59.17 | 47.43 | 59.19 | 60.45 | 73.25 | 80.62 | **80.69** |
| | CAIL-L | 75.78 | 76.35 | 78.39 | 49.38 | 58.79 | 58.21 | 49.97 | 60.34 | 58.33 | 53.87 | 56.96 | 59.90 | 75.34 | **80.88** | 78.42 |

Table 1: Overall performance on dataset CAIL small (CAIL-S) and large (CAIL-Large), where the second-best score is underlined and the best is marked with bold. The Qwen, R1, and GPT represent the Qwen2.5-7B, DeepSeek-R1, and GPT-4o, respectively.

achieving improvements of 27.16% and 22.94%, respectively, indicating the effectiveness of our method in mitigating biases in LLMs. **(3)** Our method boosts the performance of prompt-based methods. For example, on `GPT-4o`, our method achieves comparable performance with fine-tuned models, which indicates that we provide alternative solutions for users with limited resources. These observations confirm the effectiveness of our method in enhancing long-tail charge prediction.

## 4.5 ABLATION STUDY (RQ-2)

To evaluate the effectiveness of legal knowledge, we conduct an ablation study. Specifically, in our method, we first rely on the *knowledge inspector* module to learn the self-reflection knowledge, then the *knowledge integrator* retrieves candidate labels and corresponding legal rules based on the self-reflection knowledge. We first remove the self-reflection knowledge and randomly selected charges as options. This setting is denoted as *Random Options*. Secondly, we denote the options selected based on the self-reflection knowledge as *Self-reflection options*. Finally, we denote the combination of the Self-reflection Options and corresponding legal rule knowledge as *Self-reflection options+Legal rule*. We report the results of these settings in Fig. 8. It is observed that: (1) the *Self-reflection options* strategy achieves superior results compared to the *Random Options* strategy, which indicates the effectiveness of self-reflection knowledge. (2) When prompting LLMs with the self-reflection options and legal rule knowledge, the performance of small LLMs such as the Qwen2.5-7B slightly decreases. With the increment of the model size, the knowledge of legal rules improves the performance of legal reasoning. This may lie in that the larger language models process the stronger legal reasoning ability. (3) Besides, it is observed that more options result in higher performance due to the higher recall rate.

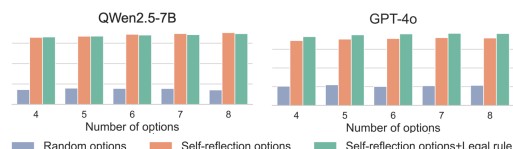

Figure 8: The results of the ablation study.

## 4.6 DOES LEGAL CASE MATTER? (RQ-3).

In this section, we examine whether incorporating legal cases enhances the legal reasoning ability of LLMs. Specifically, **(1) Legal Cases in CoT Demonstrations**. We ask the legal experts to draft legal explanations (as shown in Appendix D) for ten cases from the train set. In the inference phase, we randomly sample cases with legal explanations to elicit the LLM to think like a legal expert. As shown in Table 4, it is observed that, for smaller LLMs, the strategy of using legal cases as CoT Demonstrations reduces their legal reasoning ability, while for larger LLMs, it results in a slight but unstable improvement. **(2) Similar Cases as Demonstrations**. Following Shui et al. (2023), we prompt LLMs using similar cases retrieved by calculating the similarity between criminal facts. The results shown in Table 4 demonstrate that similar cases improve the legal reasoning ability of LLMs compared to the CoT-based methods. Although promising, more similar cases may import more noisy cases confusing LLMs and increasing the computational load.

This arises from the fact that the retriever, which relies on criminal fact similarity, struggles to distinguish confusing charges Xu et al. (2020). For example, criminal facts related to `Intentional Injury`, `Intentional Homicide`, or `Manslaughter` often contain violent actions, results of injury, or death. We report the recall rate of similar case retrievers in Table 3. It is observed that

| | | |
|---|---|---|
| **Test Cases** | | *Criminal Facts*: …,because the victim An had testified in the case of Su (already sentenced) and Zhu (already sentenced) for provoking disturbances, the defendant Su bore a grudge and at around 13:30 on June 15, 2015, he forcibly dragged An into his car …, verbally threatened and beat An while driving, causing minor injuries to An.,….
***Charge (Ground Truth)*** : Retaliating against Witnesses |
| **Zero-shot CoT (GPT-4o)** | | According to the description of the behavior, Su's behavior meets the description of the crime of Illegal Detention and Intentional Injury, in the Criminal Law of the People's Republic of China,….. In summary, Su's crime may be the crime of Illegal Detention. ✗ |
| **Few-shot CoT (GPT-4o)** | | According to the facts and legal provisions provided, we infer the possible charges against the defendant Su. **Kidnapping behavior**: The defendant Su forcibly dragged An into …; **Violence and threat behavior**: …, Su's behavior may constitute Illegal detention. ✗ |
| **RAG (GPT-4o)** | *Retrieved Case* | *Criminal Facts*: …,it was found that: the defendant An discussed with Zhang,…, to sell his sister-in-law Rishimu's one-day-old baby boy to Zhang A for RMB 50,000. Zhang A took the baby boy away and took him to the hospital for a physical examination.,…,he took the initiative to confess the above criminal facts.
***Charge (Ground Truth)*** : Trafficking in Women and Children |
| | *Answer* | Based on the facts provided, Su's behavior meets the description of illegal detention in the Criminal Law of the People's Republic of China. Illegal detention usually involves violence, …,Therefore, the crime of Su's behavior can be determined as Illegal detention. ✗ |
| **Ours (GPT-4o)** | *Legal Knowledge* | A. Provoking Trouble ; B. Smuggling Waste ; C Gathering crowds to Disturb Social Order ; D. Retaliating against Witnesses. |
| | *Answer* | A. Provoking Trouble refers to the act of …,occupying public or private property, with serious circumstances. B. Smuggling Waste refers to the act of violating customs regulations, evading customs supervision, and gaseous waste from abroad into the country. C. Gathering crowds to Disturb Social Order refers to the act of gathering a crowd to disturb social order, … D. Retaliating against Witnesses refers to the act of retaliating against witnesses,…. Therefore, the answer is: D. Retaliating against Witnesses. ✓ |

Figure 9: Case study. Examples of reasoning charge based on Zero/Few-shot CoT, RAG, and our KnowSelf based on GPT-4o. The text with a dark background represents charges.

despite an increase in recall rate with more similar cases, the LLM's reasoning outcomes did not improve as anticipated (as shown in Table 4), suggesting that additional similar cases may disrupt the model. Moreover, we analyzed 64 cases of incorrect predictions and 256 retrieved cases. It is observed that 32.2% of the retrieved cases were head cases, while 37.2% were semantically highly similar yet irrelevant cases.

### 4.7 CASE STUDY (RQ-4).

Fig. 9 shows an example of charge reasoning conducted by the framework of Zero/Few-shot CoT, RAG, and ours KnowSelf, respectively, which show the effectiveness of legal knowledge compared to the legal case-based methods. Please refer to Appendix C for more examples. We observe that the Zero/Few-shot methods conduct legal reasoning by analyzing the defendants' criminal actions and sentencing circumstances. These methods ignore the knowledge of `Retaliating against Witnesses` that the defendant's criminal behavior was to vent his dissatisfaction with the witness, which results in the misjudgment. The RAG-based frameworks conduct legal reasoning by prompting LLMs with similar cases, where similarity is calculated based on the criminal fact. However, it is challenging for retrievers to distinguish similar cases processing distinct charges when legal knowledge is lacking Xu et al. (2020). As a result, more similar cases lead to greater confusion for LLMs. Inspired by this, we propose a legal knowledge-enhanced charge reasoning framework, as shown at the bottom in Fig. 9. We retrieve legal knowledge learned by LLMs to elicit their conduct legal reasoning, which provides more reliable results.

## 5 RELATED WORKS

In this section, we briefly review three research areas related to our study, including charge prediction, legal case retrieval, and LLMs in LegalAI.

**Charge Prediction** Recently, as the foundation task of Legal AI, the charge prediction task has attracted a lot of attention achieving notable performance. For example, some work Zhong et al. (2020b); Xu et al. (2020); Zhong et al. (2018); Wang et al. (2019; 2018) improve the charge prediction performance by incorporating legal explanations such as law articles or based on the multi-task framework. Certain studies Xiao et al. (2021); Chalkidis et al. (2020) focus on fine-tuning pre-trained models tailored to the legal field. Another line of research Wei et al. (2024); Luo et al. (2023); Feng et al. (2022); Jiang et al. (2018) aims to improve the interpretability of models. Although promising, there is a limited exploration of the long-tail distribution problem in the context of charge prediction tasks, because it was commonly found in machine learning- and language modeling-based methods Mao et al. (2023). Hu et al. (2018) manually design ten distinct legal attributes for zero/few-charge prediction, despite the positive results achieved by these efforts, it limits

the model's scalability. In this study, we focus on addressing the long-tail charge prediction problem mitigating the biases learned by models.

**Legal Case Retrieval** Legal case retrieval plays a critical role in the real-world judicial scenario, which has become a research hotspot Dong et al. (2023); Li et al. (2024b; 2023). Recently, Shao et al. (2020) adopted a strategy to split the legal case into different parts and measure the similarity of these parts, which achieved promising results. Bhattacharya et al. (2022) measure the similarity between legal cases by combining the text and citation network. Recent studies Shui et al. (2023); Wei et al. (2024) show that LLMs' legal reasoning ability can be enhanced by retrieving legal cases as a demonstration, but limited by the performance of the retriever. Although legal case retrieval is considered to have great potential to improve LLM-based legal AI, its effectiveness is often unsatisfactory. For example, research has found that due to the inaccuracy of case retrieval, legal case retrieval fails to enhance retrieval-augmented generation (RAG) models and reduces the hallucinations of LLMs Karpukhin et al. (2020); Cuconasu et al. (2024). Moreover, the challenges are even greater for long-tail charge retrieval, one reason being that the proportion of relevant cases in the candidate case pool is very small. These issues lead to the failure of legal case retrieval in long-tail charge prediction, prompting us to explore methods for legal knowledge retrieval.

**LLM in Legal AI** Recently, LLMs such as GPT-4 OpenAI (2023), LLama **?**, and QWen Yang et al. (2024) have shown impressive performance on various tasks Mao et al. (2024). In the Legal AI domain, some studies explore combining LLMs with legal AI tasks. For example, Huang et al. (2023); Liu et al. (2023a); Li (2023) fine-tune LLMs on the Chinese legal corpus such as legal questions Zhong et al. (2020a) and legal case documents Deng et al. (2023). Yue et al. (2024) fine-tune LLMs to generate criminal court views. He et al. (2023) fully pre-train an LLM based on the Chinese legal dataset containing various legal AI tasks. Although these works achieve impressive performance, it is computationally expensive. To tackle this issue, some efforts explore solving legal AI tasks by prompting LLMs. For example, Yu et al. (2023; 2022); Jiang & Yang (2023) explore to design effective prompt to elicit LLM conduct legal reasoning. Shui et al. (2023); Wei et al. (2024) enhance the charge reason ability of LLMs by retrieving similar cases, which achieve promising results. However, we find that LLMs learn the legal biases due to the imbalance distribution of legal documents. To the best of our knowledge, there is a lack of research on this issue. To fill this gap, we conduct a probing study and propose a simple but effective framework KnowSelf to alleviate this problem.

## 6 CONCLUSION

In this study, we introduce a simple but effective method KnowSelf for long-tailed charge prediction. Specifically, we first check the legal knowledge learned by the LLM using the `Knowledge Inspector`. Then, we design a `Knowledge Integrator` to retrieve the knowledge that the LLM may lack. Finally, we conduct legal reasoning by prompting the LLM with the retrieved knowledge. Comprehensive experiments indicate the effectiveness and efficiency of our method.

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

## A  DETAILS OF HEAD AND TAIL CHARGES

We collect about 300 million legal cases from CJO for 187 legal charges. Table 2 summarizes several example charges and their frequency probabilities. The Fig. 10 shows the distribution of incorrect predictions of different LLMs.The X-axis denotes gold charges, and the Y-axis denotes the predicted charges. Darker circles denote a higher frequency of error prediction.

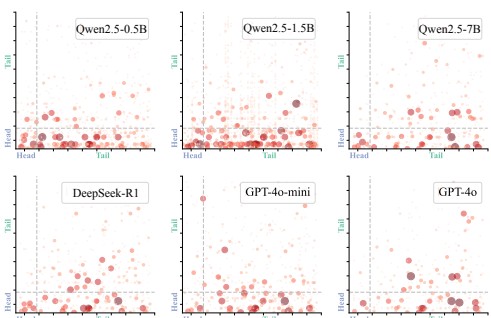

Figure 10: The distribution of incorrect predictions of different LLMs.

| Head Charges (Freq. Prob.) | | Tail Charges (Freq. Prob.) | |
|---|---|---|---|
| Theft | 19.89% | Illegal Logging | 0.69% |
| Dangerous Driving | 17.84% | Gambling | 0.66% |
| Intentional Injury | 10.42% | Intentional Destruction of Property | 0.65% |
| Traffic Accident | 8.64% | Snatching | 0.61% |
| Smuggling, Trafficking, Transporting, or Manufacturing Drugs | 6.81% | Embezzlement | 0.55% |
| Accommodating Drug Use | 3.21% | Illegal Detention | 0.52% |
| Fraud | 2.96% | Occupational Embezzlement | 0.52% |
| Provoking Trouble | 1.98% | Intentional Homicide | 0.48% |
| Robbery | 1.55% | ... | |
| Credit Card Fraud | 1.26% | Organized Looting | 0.021‰ |
| Running a Casino | 1.13% | Sabotaging Means of Transportation | 0.020‰ |
| Illegal Possession or Concealment of Firearms and Ammunition | 1.11% | Abuse of Persons Under Supervision | 0.019‰ |
| Illegal Possession of Drugs | 1.08% | Illicit Trade of Cultural Relics | 0.018‰ |
| Obstruction of Public Duties | 1.07% | High-interest Loan Reselling | 0.018‰ |
| Bribery | 0.85% | Smuggling | 0.015‰ |

Table 2: Summary of charges and their frequency probabilities

# B  REASONING WITH LEGAL CASES

To verify the effectiveness of prompting LLMs with the combination of legal cases and legal knowledge, we use the template shown in Fig. 11. The template begins with a question to ask LLM to conduct legal reasoning based on the following criminal facts and charge label pairs.

---

**Legal Knowledge + Legal Case**

You are a lawyer, please select the answer from option list based on the following criminal fact：

**Criminal Fact**: …., the defendant Ni punched Zhang Wu on the face, nose and other parts of his house at…,. It was determined that Zhang Wu's injuries were minor.

**Charge:** Intentional Injury

⋮

**Criminal Fact:** The defendant Wang and the victim Ma were sisters-in-law, …., the defendant Wang took advantage of Ma's sleep and slashed the victim's neck and stabbed her face with a knife. …, Ma's injury was grade II severe injury. According to…..

**Option List:**  **A**. Intentional Injury    **B**. Intentional Homicide    **C**. Abuse    **D**. Provoking Trouble

**Response:** The option **A** refers to acts that …; The option **B** refers to the …; The option **C** refers to physically …; The option **D** refers to acts of ….; Therefore, the answer option is:

---

Figure 11: The template of prompt LLMs for charge reasoning based on the legal cases and Knowledge.

| Retriever | Recall Rate (%) | | | | |
|---|---|---|---|---|---|
| | Top-1 | Top-2 | Top-3 | Top-4 | Top-5 |
| BM25 Shui et al. (2023) | **34.17** | **40.53** | **46.74** | **47.88** | 49.51 |
| Condenser Gao & Callan (2021) | 22.43 | 27.07 | 30.21 | 35.16 | 42.15 |
| coCondenser Gao & Callan (2022) | 22.67 | 29.88 | 32.40 | 34.53 | 44.23 |
| SEED Lu et al. (2021) | 25.55 | 31.60 | 38.77 | 44.60 | 48.76 |
| RetroMAE Liu et al. (2023c) | 29.97 | 33.93 | 35.10 | 44.14 | **49.70** |

Table 3: The performance of similar case retrieval.

| # Cases | Models' Perfomance of F1-Scores (%) | | | | |
|---|---|---|---|---|---|
| | Qwen2.5-1.5B | Qwen2.5-7B | BaiChuan2-13B | GPT-4o-mini | GPT-4o |
| *Injecting N(1~4) Legal Cases in CoT Demonstrations* | | | | | |
| 0-shot | **25.77** | **44.64** | **38.16** | 30.28 | 46.20 |
| 1-shot | 22.78(-2.99) | 35.98(-8.66) | 37.26(-0.90) | 30.58(+0.30) | 44.82(-1.83) |
| 2-shot | 24.46(-1.31) | 29.96(-14.68) | 37.16(-1.00) | 29.18(-1.10) | **46.24(+0.04)** |
| 3-shot | 24.06(-1.71) | 32.6(-12.04) | 34.53(-3.63) | **31.15(+0.87)** | 45.62(-0.58) |
| 4-shot | 24.39(-1.38) | 36.59(-8.05) | 34.62(-3.54) | 34.62(+4.34) | 45.33(-0.87) |
| *Retrieving N(1~4) Similar Cases as Demonstrations* | | | | | |
| 1-shot | 40.10 | 47.91 | 49.11 | 42.6 | 55.11 |
| 2-shot | 40.33(+0.23) | **48.00(+0.09)** | **49.28(+0.17)** | 43.85(+1.25) | 54.8(-0.31) |
| 3-shot | 39.82(-0.28) | 46.16(-1.75) | 47.69(-1.42) | 45.08(+2.48) | **55.48(+0.37)** |
| 4-shot | **40.5(+0.4)** | 45.72(-2.19) | 45.99(-3.12) | **45.75(3.15)** | 55.06(-0.05) |

Table 4: The performance of prompting LLMs using legal cases as demonstrations.

## C    MORE EXAMPLES OF LEGAL REASONING

In this section, in order to demonstrate the effectiveness of our method, we provide more examples of the results generated based on the baseline methods and KnowTai in Table. 5.

| | |
|---|---|
| Fact-1 | The victims Gao and Zhang are the parents of the defendant. The three of them have lived together in the same courtyard for a long time. In family life, the defendant often beats and scolds his parents after drinking because of trivial matters. At about 12:00 on March 20, 2017, the defendant had a dispute with Gao and Zhang over trivial family matters at his home. Later, the defendant beat Gao and Zhang with bricks and kettles, causing soft tissue contusion on Gao 2's right shoulder blade and contusion on Zhang's left upper arm. Gao and Zhang were forced to leave after being injured, so the defendant chased them on a motorcycle and threatened the two not to call the police. After the defendant returned home, he smashed the rice cooker, TV, refrigerator, iron pot, jar, and other items in his parents' house, ... |

| | |
|---|---|
| Zero-CoT | Based on the facts and evidence provided, it can be inferred that the defendant may be involved in the following crimes: 1. Intentional injury: The defendant had a dispute with his parents over trivial family matters, and beat his parents with bricks and kettles, causing minor injuries to his parents. His behavior meets the characteristics of intentional injury. According to Article 234 of the Criminal Law of the People's Republic of China, acts of intentionally injuring the body of others should be deemed as intentional injury. 2. Intentional destruction of property: After returning home, the defendant smashed the rice cooker, television, refrigerator, iron pot, jar and other items in his parents' room, with a value of RMB 1,049.00, which meets the requirements of the crime of intentional destruction of property. According to Article 275 of the Criminal Law of the People's Republic of China, intentional destruction of public or private property in a large amount constitutes the crime of intentional destruction of property. In summary, based on the facts and relevant legal provisions, the defendant's behavior may constitute intentional injury and intentional destruction of property. |
| Few-CoT | Based on the facts and evidence provided, it can be inferred that the defendant may be involved in the following crimes: 1. intentional injury: the defendant had a dispute with his parents over trivial family matters, and beat his parents with bricks and kettles, causing minor injuries to his parents. His behavior meets the characteristics of intentional injury. According to Article 234 of the Criminal Law of the People's Republic of China, the act of intentionally injuring another person's body should be deemed as intentional injury . 2. Intentional destruction of property: After returning home, the defendant smashed the rice cooker, TV, refrigerator, iron pot, jar, and other items in his parent's room, with a value of RMB 1,049.00, which meets the requirements of the crime of intentional destruction of property. According to Article 275 of the Criminal Law of the People's Republic of China, intentional destruction of public or private property in large amounts constitutes the crime of intentional destruction of property.In summary, based on the facts and relevant legal provisions, the defendant the defendant's behavior may constitute the crime of intentional injury and the crime of intentional destruction of property. |
| RAG (Retrieved Case) | At about 1:00 a.m. on November 19, 2015, the defendant Yang 1 was drinking with Gao 1 and "Aliang" (homonymous, the latter two are now at large) at the night snack stand in front of Henghui Auto Shop on Longjiang Farm Commercial Street in Baisha County. Gao 1 recognized Zhang and Zhang A, who had had a conflict with him at the next table. Gao 1 then called Gao 2 and Fu 2 (homonymous, the latter two are now at large) to come over and beat Zhang and Zhang A together. When Gao 2 and Fu 2 arrived on a motorcycle, Zhang and Zhang A were just sitting on the motorcycle and preparing to leave. Gao 2 stepped forward and chopped Zhang's head and back with a machete he was carrying, knocking Zhang to the ground. Seeing this, Zhang A jumped out of the car and ran away. Gao 1 took the machete from Gao 2 and chased Zhang A. Yang 1, Gao 2, "Aliang", Fu 2 and others also followed suit and chased Zhang A. Yang 1 and others caught up with and beat Zhang A at the Longyuan optional shopping mall. Zeng, who was having a midnight snack with Zhang and Zhang A, ran over to stop the fight. Zhang A was able to break free and ran back to the midnight snack stall. Yang 1 and others chased Zhang A back and continued to beat him with knives, glass bottles and other objects, but was stopped by Zeng, and Yang 1 and others left the scene. **Charge: Intentional injury; provoking trouble** |

| | |
|---|---|
| RAG (Answers) | According to the facts and legal provisions provided, the following charges can be considered:1. Intentional injury:" The defendant beat his parents with bricks and kettles, causing minor injuries to the two, which meets the constituent elements of intentional injury. Although the injury was assessed as minor, it still falls within the scope of intentional injury. 2. Intentional destruction of property:, the defendant smashed rice cookers, TVs, and other items after returning home. This behavior meets the constituent elements of the crime of intentional destruction of property. Although the total value of the damaged items is 1,049 yuan, which is relatively small, it still constitutes the crime of intentional destruction of property.In summary, based on the analysis of the facts and evidence provided, the defendant may be suspected of intentional injury and intentional destruction of property in this case. |
| Ours | **Knowledge: A. Abuse B. Abuse of a person under custody:**
A. Abuse refers to the act of physically and mentally torturing and persecuting family members who live together by beating, scolding, tying up, freezing, starving, restricting freedom, humiliating personality, denying medical treatment, or forcing excessive labor, etc., with serious circumstances The public prosecutor's criminal charges against the defendant Li Mouyun are clear in facts, and the evidence is solid and sufficient. The charges are established and should be supported. B. Abuse of supervised persons refers to the act of supervisors in prisons, detention centers, and other regulatory agencies beating or physically abusing supervised persons, with serious circumstances.Therefore, the answer is: A. Abuse |
| Fact-2 | The defendant Yu was a drug addict who had hallucinations and paranoid delusions due to drug use and therefore committed intentional harm to others. On the evening of November 22, 2013, the defendant Yu again took drugs in Tianxin District, Changsha, and had hallucinations. At about 10 am the next day, near Wuguang South Station in Changsha, he boarded a minibus for Changsha West Bus Station. At about 10:30 on the same day, when the bus was driving near the west of Houzishi Bridge in Yuelu District, Changsha, the defendant Yu Mou continued to have hallucinations of being chased by people due to drug use and was extremely nervous. He took a butcher knife and a skinning knife he carried with him and drove all the passengers out of the minibus. Then, he hijacked the driver Zou Mou, and the conductor Bao Mou with a knife and forced the driver to drive him to a safe place. When the vehicle hijacked by the defendant Yu Mou passed the Meixi Lake section of the West Second Ring Road in Yuelu District, Changsha City, he was captured by the public security organs. ... |

| | |
|---|---|
| Zero-CoT | Based on the above facts, the defendant Yu's behavior may involve the following crimes: 1. Car hijacking: Yu drove the passengers out of the minibus with a knife, hijacked the driver and the conductor, and asked the driver to take him to a safe place, which constituted car hijacking. According to the description of the facts, it can be determined that he controlled the car and affected the normal driving and safety of the vehicle. 2. Intentional injury: Although there is no mention of Yu physically hurting others in this specific incident, his past hallucinations and delusions caused by drug use may be considered in sentencing. 3. Illegal possession of cold weapons: The butcher knife and skinning knife carried by Yu may constitute illegal possession or carrying of controlled instruments. It should be noted that, in view of the judicial appraisal opinion, Yu was in the onset of mental illness when committing the crime, which may affect her criminal responsibility. This situation may be considered as a factor for mitigation or exemption of punishment during sentencing, depending on the judicial determination of her capacity for responsibility at the time. In judicial practice, if a person is completely incapable of criminal responsibility, he may not be held criminally liable, while if his capacity for criminal responsibility is limited, the punishment may be mitigated. In summary, Yu's main crime is likely to be car hijacking, but the determination of his criminal responsibility will play a key role in the trial. |
| Few-CoT | According to the description of the case and relevant evidence, the following is a step-by-step reasoning and identification process of the defendant Yu's behavior: 1. Basic facts: The defendant Yu is a drug addict, and he has hallucinations and delusions of persecution due to drug use. Yu held a knife to hijack the driver and the conductor, and forced the driver to drive until he was arrested by the public security organs. 2. Analysis of legal facts: Yu held a knife to force passengers to get off the bus and hijacked the driver and the conductor, which was a behavior of hijacking others by violence or other methods. Yu's behavior forcibly deprived the driver and the conductor of their personal freedom, posing a threat to the personal safety of passengers and people in the car.3. Correspondence analysis with criminal law provisions: According to Article 239 of the Criminal Law of the People's Republic of China, the crime of kidnapping refers to the crime of kidnapping others as hostages by detention, hijacking or other coercive methods, so as to force a third party to do a certain act. The defendant Yu's behavior meets the characteristics of hijacking others, and at the same time, it is carried out in a violent manner. Although it is based on his hallucinations after taking drugs, it cannot exempt him from criminal responsibility. 4. Analysis of mental state and responsibility: Although the appraisal shows that Yu has a mental disorder caused by drug abuse and was in the onset of mental illness when committing the crime, according to the law, this situation cannot completely exempt him from criminal responsibility, especially in the case of voluntary drug use. In summary, combined with the case and legal provisions, the defendant Yu's behavior constitutes the crime of kidnapping. |

|  |  |
|---|---|
| RAG (Retrieved Case) | In November 2013, the criminal Yu (already sentenced) learned from the defendant Li that Zhou (handled in another case) had a gun during a chat, so he asked the defendant Li to contact Zhou to purchase the gun. On January 1, 2014, the defendant Li and Zhou agreed on the price of the pistol at RMB 8,000. On the evening of January 4, 2014, Zhou arranged for the criminal Xu (already sentenced) to take a homemade black pistol and six bullets from Qingyuan City, Guangdong Province to Changsha City, Hunan Province to meet with Yu and the defendant Li for the transaction. In the early morning of January 5, 2015, Xu arrived in Changsha. After failing to contact the defendant Li, Xu returned to his hometown in Xiangyin, Hunan with the gun and bullets. Shortly after arriving home, Xu received a call from Zhou, asking him to deliver the pistol and bullets to the Haitian Building in Yuelu District, Changsha City. Xu then contacted the criminal Yang (already sentenced) and asked him to drive him to Haitian Building, and Yang agreed. On the afternoon of January 5, 2014, Xu took a black BYD car with license plate number Hunan F×××× driven by Yang to the entrance of the Haitian Building in Changsha, and met with Yu and defendant Li. Yu said it was unsafe to test the gun at the entrance of Haitian Building, so he drove a white off-road vehicle with license plate number Hunan A××××× and took defendant Li to lead the way. Yang took Xu to follow behind and drove to the construction site of "Xindamei" real estate project in Meixi Lake, Yuelu District, Changsha. Yu, Xu, and defendant Li got out of the car to test the gun at the construction site. Yu and Xu tried the guns one after another, and they jammed after firing and could not be used normally. You immediately said he did not want the gun. Defendant Li contacted Zhou, who agreed to change the gun and entrusted Defendant Li to keep the pistol and the remaining bullets on his behalf. Zhou also asked defendant Li to give Xu 1,000 yuan for travel expenses. Xu handed the homemade pistol and bullets to the defendant Li and placed them in the storage compartment of the co-pilot's white off-road vehicle with license plate number Xiang A×××××. Yu gave Xu 1,000 yuan. Afterward, Xu and Yang drove back to Xiangyin County. On the evening of January 6, 2014, Zhou contacted Xu to take the pistol and bullets back to Qingyuan, Guangdong. Xu rented Yang's car and met with Yu and the defendant Li at Haitian Building in Yuelu District, Changsha City. The defendant Li handed the pistol and bullets to Xu from Yu's car. At about 22:30 that night, when Yang was driving Xu through the Shahe Police Station on Furong North Road in Changsha City, he was arrested by the police on duty. A black pistol and four bullets were found on the pedal of the right rear seat of his car. **Charge: Illegal manufacture, sale, transportation, mailing, and storage of firearms, ammunition, and explosives** |
| RAG (Answer) | According to the facts and legal provisions provided,the following charges can be considered:1. Intentional injury:" the defendant beat his parents with bricks and kettles, causing minor injuries to the two, which meets the constituent elements of intentional injury. Although the injury was assessed as minor, it still falls within the scope of intentional injury. 2. Intentional destruction of property:, the defendant smashed rice cookers, TVs and other items after returning home. This behavior meets the constituent elements of the crime of intentional destruction of property. Although the total value of the damaged items is 1,049 yuan, which is relatively small, it still constitutes the crime of intentional destruction of property.In summary, based on the analysis of the facts and evidence provided, the defendant may be suspected of intentional injury and intentional destruction of property in this case. |

| | |
|---|---|
| Ours | **Knowledge: A. Kidnapping B. Waste smuggling C. Illegal detention D. Hijacking of ships and cars**
A. Kidnapping refers to the act of kidnapping someone for the purpose of extorting money or holding someone hostage. B. Waste smuggling refers to the act of violating customs regulations, evading customs supervision, and transporting solid waste, liquid waste, and gaseous waste from abroad into the country, with serious circumstances. C. Illegal detention refers to the act of illegally detaining others or illegally depriving others of their personal freedom by other means. D.Hijacking a ship or a car refers to the act of hijacking a ship or a car by violence, coercion, or other means. Therefore, the answer is: D. Hijacking a ship or a car . |
| Fact-3 | ...,Xiao and the defendant Song had a debt dispute. On March 26, 2013, Luo bought 2,260 laying ducks from Xiao for 32,000 yuan and paid the full amount on the 28th of the same month. At about 13:00 on April 2, 2013, the defendant Song gathered Zhao, Song 2 and more than a dozen others on the pretext of taking back the laying ducks Xiao sold to Luo to pay off the debt. They drove to Luo's duck farm located at Luoyingtang Reservoir, Weixin Village, Dongjin Town, Gangnan District, Guigang City, and robbed 2,253 laying ducks worth 37,700 yuan despite Luo's dissuasion and beating Luo. After the incident, the public security organs seized the 2,253 laying ducks and returned them to Luo. ... |
| Zero-CoT | According to the case you provided, I will gradually infer the relevant charges: 1. Elements of the crime: Behavior: Song went to the duck farm with more than a dozen people, forcibly seized and transported 2,253 laying ducks. Object: Property belonging to others, specifically laying ducks. Subjective aspect: Intentional illegal possession, although there is a debt dispute, it is obtained by robbery, rather than resolved through legal channels. Result: Direct economic losses caused to others (although the ducks have been recovered since then, the behavior during the robbery has constituted infringement). Legal analysis This case involves the act of illegally seizing other people's property, which legally meets the elements of the crime of robbery. The defendant Song gathered others to jointly commit violent acts and forcibly took the legal property owned by outsiders for himself, and the amount was huge. 3. Factors for lenient or reduced punishment After the incident, Song took the initiative to surrender and truthfully confessed his criminal behavior. This is a self-surrendering behavior, which can be legally lenient or reduced. The defendant actively compensated the victim for the loss and obtained the victim's forgiveness, and requested a lenient punishment. In summary, based on the above facts and analysis, the defendant Song's behavior meets the constituent elements of the crime of robbery. Combined with his surrender and compensation, it can be recommended that he be given a lighter punishment. Therefore, the relevant crime should be "robbery". |

Few-CoT

In the case, the defendant Song , on the pretext of taking back the egg-laying duck to pay off debts, gathered others and used violence to rob the egg-laying duck. Next, I will reason step by step and analyze the possible crimes that Song may have committed: 1. The nature of the behavior: Song, on the pretext of resolving a debt dispute, gathered many people and robbed the egg-laying duck from the victim Luo 1 by means of intimidation and violence. This behavior belongs to the use of force or threats to forcibly seize property for the purpose of illegal possession. 2. The object and value of the robbery: The object of the robbery is the egg-laying duck, with a total value of 37,700 yuan, and it was taken by violence, which is consistent with the description of the crime of robbery in my country's Criminal Law. 3. The conformity of specific behavior with legal provisions: According to Article 263 of my country's Criminal Law, the crime of robbery refers to the act of forcibly robbing public or private property by violence, coercion or other methods for the purpose of illegal possession. Song Mou's behavior meets this point. 4. Subsequent behavior: Song took the initiative to surrender after the incident and truthfully confessed the crime, which is a self-surrender. In addition, he voluntarily compensated the victim for the corresponding economic losses and obtained the victim's forgiveness. Based on the above analysis, Song's behavior constitutes robbery.

| | |
|---|---|
| RAG
(Retrieved Case) | On March 25, 2002, the defendant Luo 4 gathered six or seven people to beat up the driver Chen 2 and Chen 6, father and son, on the pretext that he was splashed with water by a coal truck on Jiang 1's motorcycle to Sha. Chen 2's injury was identified as a minor injury. On the afternoon of August 18, 2003, Jiang 1's (already sentenced) younger brother Jiang 1 was scratched by Luo 7's motorcycle. Luo 7 called Luo 1 to act as an intermediary to handle the matter. Luo 1 said that 10 yuan would be enough. Jiang 1 was dissatisfied and called Jiang 1 to tell him about it. Jiang 1 gathered Luo 4, Ma and others to chop Luo 1 with kitchen knives. It was identified that Luo 1's injury constituted a minor injury. On the afternoon of April 20, 2004, Wu 4 from Wenquan Village, Jinjiang Town, Linwu County, had an argument and fight with Tan 1 Zhong and Tan 1 Hua from Gui County over a debt dispute, and Wu 4 injured Tan 1 Hua. Tan 1 Zhong helped Tan 1 Hua to the infirmary of the Jinjiang Town Secondary Power Station for treatment. After learning about the situation, Wu 4's nephew Wu 1 (on the run) gathered the defendants Luo 4 and Jiang 1 Jian and others to rush to the Jinjiang Town Secondary Power Station, and used wooden sticks to hit Tan 1 Zhong's back, waist, abdomen, legs, etc., causing Tan 1 Zhong to be seriously injured (fifth degree disability). One day in May 2005, Jiang Jian gathered more than ten people including the defendant Luo and He (who have been sentenced) to Cao Kai's home in Guya Chong Village, and demanded Cao Kai to pay more than 10,000 yuan on the grounds that the Sha Police Station had seized the TV set in Jiang Jian's home, which was caused by Cao Kai. Cao Kai refused, so Jiang Jian asked Luo and others who went with him to beat Cao Kai's son Cao. Later, Cao Kai was forced to "compensate" Jiang Jian, Luo and others for 12,700 yuan. At about 5:00 p.m. on April 2, 2006, the defendant Luo 4 had an argument with Tang 1 from Wen 1 Chong Village in an Internet cafe in Shitoumiao Village, Linsen Village Committee, Jinjiang Town, Linwu County, over trivial matters. Luo 4 was beaten by people from Wen 1 Chong Village and felt that he had lost face. The next day, he gathered Yan 5, Yan 3 (also known as Yan 1 Zeng, who has been sentenced), Xiao 3 (also known as Xiao 1, who has been sentenced), Lei 5 (who has been sentenced) and others to detain Tang 12 and Tang 7 from Wen 1 Chong Village and take them to the side of the road at 199km+80m of Provincial Highway S214 (near Heye Toll Station, Gui County). Luo 4 instructed Sun 1, Yan 1 Zeng and others to drag Tang 12 and Tang 7 out of the car and cut their hands and tendons. According to the appraisal, the victim Tang 12's injuries constituted minor injuries, and Tang 7's injuries constituted serious injuries (eighth degree disability).
**Charge: Intentional injury; kidnapping; extortion** |
| RAG
(Answer) | "According to the facts and legal provisions provided,the following charges can be considered:1. Intentional injury:" the defendant beat his parents with bricks and kettles, causing minor injuries to the two, which meets the constituent elements of intentional injury. Although the injury was assessed as minor, it still falls within the scope of intentional injury. 2. Intentional destruction of property:, the defendant smashed rice cookers, TVs and other items after returning home. This behavior meets the constituent elements of the crime of intentional destruction of property. Although the total value of the damaged items is 1,049 yuan, which is relatively small, it still constitutes the crime of intentional destruction of property.In summary, based on the analysis of the facts and evidence provided, the defendant may be suspected of intentional injury and intentional destruction of property in this case. " |

| | |
|---|---|
| Ours | **Knowledge: A. Robbery B. Forced labor C. Looting by a mob D. Illegal entry into a residence**
A. Robbery refers to the act of robbing public or private property with the purpose of illegal possession by using violence, coercion or other methods. B. Forced labor refers to the act of forcing others to work by means of violence, threats or restriction of personal freedom. C. Looting by a mob refers to the act of gathering a large number of people to openly rob public or private property in large amounts or with serious circumstances. D. Illegal entry into a residence refers to the act of illegally and forcibly breaking into another person's residence without the consent of the owner of the residence, or refusing to leave after being asked to leave. Therefore, the answer is: C. Looting by a mob . |

Table 5: Examples of charge reasoning based on Zero/Few-shot, RAG and our framework

## D  FACTS AND LEGAL EXPLANATIONS

In this section, we will demonstrate some examples of the facts and legal explanations given by experts, which are shown in Table 6.

| | |
|---|---|
| Fact | On November 29, 2017, Li sneaked into Wu's residence and stole a 4K laptop computer worth RMB 7,184. The stolen property has been recovered and returned to the victim Wu. ... |
| Legal explanation | The defendant Li broke into other people's homes for the purpose of illegal possession and stole a large amount of property. His behavior constituted the crime of theft. |
| Fact | In December 2017, Zhu fabricated a scam to the victim An about making money by jointly speculating in real estate. He lied that he and the victim An jointly invested in the purchase of property B for the purpose of speculation and appreciation. The victim An then transferred RMB 250,000 to Zhu. After receiving the money, Zhu used 250,000 yuan to repay debts and for personal consumption. ... |
| Legal explanation | The defendant Zhu ignored national laws, fabricated facts, concealed the truth, and defrauded others of property in a huge amount. His actions constituted the crime of fraud. |
| Fact | In the early morning of March 9, 2018, public security police arrested Tang in the courtyard of the Collection Bureau directly under the Qianjiang City Local Taxation Bureau and found a bag of methamphetamine crystals (ice, net weight 49.75 grams) in a transparent plastic bag in his shirt pocket. ... |
| Legal explanation | The defendant Tang violated the state's drug management system and illegally possessed 49.75 grams of methamphetamine. His behavior constituted the crime of illegal possession of drugs and the circumstances were serious. |
| Fact | At about 23:00 on January 6, 2019, Xia and Xiang had a conflict over trivial matters, so they asked each other to meet and resolve it. Later, when Xia brought a kitchen knife to the sidewalk in front of the Yunliang Jinquan Hotel at the intersection of Panlong District in this city, he met Zhang, Xiang and Wang (both handled in separate cases) who were carrying steel pipes. The two sides fought each other. Xiang and Wang beat Xia on the head and caused minor injuries. Xia swung with a kitchen knife and caused Xiang Mou to suffer a second-degree minor injury and Wang to suffer a minor injury. On January 7, 2019, Xia voluntarily surrendered to the Lishutou Police Station. It was also found that Xi, who participated in the fight after the incident, had a mutual understanding with Xiang and Wang. ... |
| Legal explanation | The defendant Xia ignored the national laws, invited others to fight with weapons due to trivial disputes, and his behavior constituted the crime of gathering a crowd to fight. |

Table 6: Examples of facts and legal explanations

