# OpenReview forum: "Revealing and Pushing Legal Reasoning Boundary of Large Language Models"
_ICLR.cc/2026/Conference — ICLR 2026 Conference Withdrawn Submission_

### Official Review · Reviewer_d4Vw · 2025-10-20

**Soundness:** 2
**Presentation:** 2
**Contribution:** 1
**Rating:** 4
**Confidence:** 4

**Summary:**

This paper introduces KnowSelf framework, which consists of a Knowledge Inspector to identify their knowledge gaps and a Knowledge Integrator to provide tailored legal knowledge for accurate legal reasoning. It focuses on the long-tailed situation in the legal field. The experiment on the charge prediction task on CAIL dataset shows the effectiveness.

**Strengths:**

1. This paper studies the long-tail problem in the legal reasoning field, which is realistic.
2. The analysis of the long-tail problem is interesting, providing insights for further studies.

**Weaknesses:**

1. **There is a problem in the reasoning part of the method.** The true label may not necessarily be included in the given options. In such cases, according to the provided prompt template, the model would never produce the correct answer. I would like to know how the authors handle this situation. Moreover, if the label is directly mixed into the options for inference, it introduces **label leakage**, which compromises the rigor of the experiments.

2. **In Section 4.5 (Ablation Study),** it is unclear **where the “Legal rule” comes from**, as it does not seem to be mentioned in the preceding methodological description.

3. **The experimental results appear to lack consistency.** In the CAIL-S dataset, the KnowSelf method outperforms GPT in both head and tail classes (R1 metric), yet GPT achieves a higher overall score (ALL). Could the authors clarify this discrepancy?

4. **Regarding the definition of head and tail cases,** why was the threshold set at **80%**? Is this value supported by any prior literature?

5. **Typos and citation issues:**

   * Line 194: *“the Knowledge Integrator module retrieves the missing legal knowledge according to the findings of the Knowledge Integrator module”* — the phrase “according to the findings of the **Knowledge Integrator** module” seems redundant or incorrect.
   * In **Table 1**, “GAG” appears to be a typo.
   * Line 449: “LLama” seems to be improperly cited.

6. **Lack of related baselines.** There exist many legal-domain models [1-3] that have been shown to perform effectively on the CAIL task. These models should be included for discussion or empirical comparison to better demonstrate the advantages and limitations of the proposed approach.


***

[1] Chen, H., Xu, Y., Wang, B., Zhao, C., Han, X., Wang, F., ... & Xu, Y. (2025). LexPro-1.0 Technical Report. arXiv preprint arXiv:2503.06949.

[2]  Zhou, Z., Yu, K. Y., Tian, S. Y., Yang, X. W., Shi, J. X., Song, P., ... & Li, Y. F. (2025). Lawgpt: Knowledge-guided data generation and its application to legal llm. arXiv preprint arXiv:2502.06572.

[3] W. He et al. Hanfei-1.0. github, 2023

**Questions:**

See weaknesses above

---

### Official Review · Reviewer_FUs4 · 2025-10-20

**Soundness:** 1
**Presentation:** 1
**Contribution:** 2
**Rating:** 0
**Confidence:** 5

**Summary:**

The paper presents an approach called KnowSelf, aimed at addressing the long-tail charge prediction problem in legal reasoning using large language models (LLMs). The authors argue that existing LLMs struggle with this issue due to the imbalanced distribution of legal data during pre-training, which affects their inference capabilities, particularly for infrequent charges. KnowSelf incorporates a Knowledge Inspector to identify knowledge gaps in the LLMs and a Knowledge Integrator to provide tailored legal knowledge, improving the accuracy of charge predictions.

**Strengths:**

The problem itself is important.

**Weaknesses:**

1. The paper is poorly written, with important figures put in Appendix (e.g., the authors spend a whole paragraph in the main text describing Fig. 10 that is in the appendix) and some sentences incomplete (”option from the following list.”, ”We depict the table T in Fig. 10.”). The title is even different in the pdf and openreview.
2. The novelty of the paper is limited as the whole framework is similar to [1]. The main difference is that [1] trains a small model as the knowledge inspector while the authors use LLM reflections on the training data to build it.
3. Many design problems. Particularly, the authors claim that legal case retrieval cannot help long-tail LJP, but they only use general retrievers to retrieve legal case, which contradicts to their motivation. There have already been many SOTA legal case retrievers that retrieve documents beyond text similarities [2,3,4]. None of them is used in the proposed method or tested as baselines.

[1]Li, H., Ai, Q., Chen, J., Dong, Q., Wu, Z. and Liu, Y., 2025, April. Blade: Enhancing black-box large language models with small domain-specific models. In *Proceedings of the AAAI Conference on Artificial Intelligence* (Vol. 39, No. 23, pp. 24422-24430).

[2]Li, H., Ai, Q., Han, X., Chen, J., Dong, Q. and Liu, Y., 2025, April. Delta: Pre-train a discriminative encoder for legal case retrieval via structural word alignment. In *Proceedings of the AAAI Conference on Artificial Intelligence* (Vol. 39, No. 25, pp. 27072-27080).

[3]Su, W., Ai, Q., Wu, Y., Xie, A., Wang, C., Ma, Y., Li, H., Wu, Z., Liu, Y. and Zhang, M., 2025. Pre-training for Legal Case Retrieval Based on Inter-Case Distinctions. *ACM Transactions on Information Systems*, *43*(5), pp.1-27.

[4]Li, H., Ai, Q., Chen, J., Dong, Q., Wu, Y., Liu, Y., Chen, C. and Tian, Q., 2023, July. SAILER: structure-aware pre-trained language model for legal case retrieval. In *Proceedings of the 46th International ACM SIGIR Conference on Research and Development in Information Retrieval* (pp. 1035-1044).

**Questions:**

What’s GAG, Self-Consistency in Table 1?

---

### Official Review · Reviewer_XnxD · 2025-10-31

**Soundness:** 2
**Presentation:** 3
**Contribution:** 2
**Rating:** 2
**Confidence:** 4

**Summary:**

This paper proposes KnowSelf, a modular framework for enhancing legal reasoning of large language models (LLMs), especially for long-tail charge prediction tasks.
The framework introduces a Knowledge Inspector that identifies the model’s knowledge deficiencies by analyzing confusion pairs, and a Knowledge Integrator that retrieves relevant legal knowledge (definitions, statutes) to supplement reasoning. Experiments are conducted on CAIL datasets using several LLMs such as GPT-4o, Qwen2.5-7B, and DeepSeek.

**Strengths:**

The paper is very clearly written and easy to follow. The motivation, model design, and experimental setup are logically organized and visually well-presented (especially Figures 2–4).

The work addresses a real and important problem in LegalAI—handling long-tail charges, which is a persistent weakness for existing LLMs.

The proposed method yields consistent improvements across models and datasets, demonstrating that the self-reflective retrieval approach can indeed mitigate some common confusion errors.

**Weaknesses:**

1. While the idea of using confusion analysis to guide retrieval is sensible, it feels incremental. The paper essentially combines well-known ideas — confusion matrix analysis, retrieval-augmented reasoning, and prompt-based legal knowledge injection — under a new framework name.
There is no substantial algorithmic or theoretical innovation beyond this integration.

2. The “Knowledge Inspector” mainly counts prediction errors and selects frequently confused labels. This is a straightforward heuristic rather than a genuinely learned or adaptive mechanism. No attempt is made to model uncertainty, causal confusion, or representation-level introspection.

3. Only two datasets (CAIL-Small/Large) are used, both from the same legal corpus and language (Chinese criminal law). It’s unclear if the method generalizes to other jurisdictions or legal tasks.  Ablation studies are minimal and mostly qualitative; more systematic quantitative breakdowns (e.g., per-category recall, noise sensitivity) would strengthen the claim.

4. The method requires precomputing a full confusion matrix using labeled data, which may not be feasible for low-resource or unseen tasks. The paper does not discuss the computational or annotation cost of building the reflection table.

**Questions:**

none

---

### Official Review · Reviewer_i5Nd · 2025-10-31

**Soundness:** 2
**Presentation:** 2
**Contribution:** 2
**Rating:** 2
**Confidence:** 4

**Summary:**

The paper addresses the task of long-tail legal charge prediction. The authors propose identifying which frequently occurring charges tend to mask rare (tail) charges for a given model, and incorporating this information in-context during inference. This approach leads to substantial improvements over zero-shot, RAG, and (weak) fine-tuning baselines.

**Strengths:**

* The overall presentation is clear.
* The method leads to large improvements in performance compared to zero-shot and RAG.
* I appreciate the ablation showing that rule retrieval is more effective than case retrieval.

**Weaknesses:**

The authors show that the proposed method yields substantial improvements over standard RAG. However, it relies on the availability of a training set, which raises the question of how well simply fine-tuning on that same train set would perform. The authors include some fine-tuning baselines, but these are quite weak: they fine-tune very old models (e.g., released in 2021 or earlier). This, in my view, is the main limitation of the work. Even fine-tuning such outdated models achieves performance surprisingly close to their method (see Table 1), which suggests that modern models might surpass KnowSelf. For a fairer assessment, the authors should fine-tune recent LLMs. If computational constraints are a concern, at minimum they could fine-tune small models such as SmolLM2 360M or Qwen 3 0.6B, for example following a setup like [1], but the 8B range would be preferable.
* Results are presented only for a charge prediction problem. There are many legal classification tasks with long tails, and the paper would be stronger by showing results in other such (legal) tasks.
* Writing needs additional work.

[1] Dominguez-Olmedo, R., Nanda, V., Abebe, R., Bechtold, S., Engel, C., Frankenreiter, J., ... & Livermore, M. (2025). Lawma: The Power of Specialization for Legal Annotation. In The Thirteenth International Conference on Learning Representations (pp. 1-45). International Conference on Learning Representations.

**Questions:**

Notes:
 * My knowledge of RAG approaches is too limited to judge technical novelty.
 * The title of the manuscript does not match that of OpenReview.

Questions:
* L312: isn’t it more appropriate to report the median/mean rather than the best performance? It would also be useful to include all seed results in the appendix.
* L307: did you perform ablations to ensure 10 epochs is not too many? Performance may degrade by that time.
* L289: so the total train size is 187 * 10 = 1870?
* L351: is the correct choice always among the choices presented to the model? Or only if sampled by chance?
 * L216: what temperature is used for sampling?
 * L147: “LLM-based methods have become mainstream for charge prediction” - do you know of LLM-based systems being used in practice for charge prediction?

Comments:
 * Please make the figures larger, they are currently not readable in print. To make space, you could move 4.7 and Figure 9 to the Appendix.
* Fix the citations, e.g., using \citet or \citep as needed
 * L254: the text is cut-off.
* L360: it may also be that the smaller models provide inaccurate definitions of the charge labels.
* L449: Llama ?

**Details Of Ethics Concerns:**

The paper uses LLMs to infer criminal charges based on the description of a legal case. I think that this is a potentially sensitive application domain.

---

### Note · Authors · 2025-11-29

I have read and agree with the venue's withdrawal policy on behalf of myself and my co-authors.